# Training Large Reasoning Models Efficiently via Progressive Thought Encoding

**Zeliang Zhang**[1][†][*]**, Xiaodong Liu**[2][†]**, Hao Cheng**[2]**, Hao Sun**[2]**, Chenliang Xu**[1] **and Jianfeng Gao**[2]
[1]University of Rochester, [2]Microsoft Research

## Abstract

Large reasoning models (LRMs) excel on complex problems but face a critical barrier to efficiency: reinforcement learning (RL) training requires long rollouts for outcome-based rewards, where autoregressive decoding dominates time and memory usage. While sliding-window cache strategies can bound memory, they disrupt long-context reasoning and degrade performance. We introduce *Progressive Thought Encoding*, a parameter-efficient fine-tuning method that enables LRMs to reason effectively under fixed-size caches. By progressively encoding intermediate reasoning into fixed-size vector representations, our approach eliminates the need to backpropagate through full-cache rollouts, thereby reducing memory usage, while maintaining constant memory during inference. Experiments on three models, including `Qwen2.5-3B-Instruct`, `Qwen2.5-7B-Instruct`, and `DeepSeek-R1-Distill-Llama-8B`, on six widely used challenging mathematical benchmarks show consistent gains: our method achieves +19.3% improvement over LoRA-based fine-tuning and +29.9% over LRMs without fine-tuning on average, with up to +23.4 accuracy improvement on AIME2024/2025 under the same tight cache budgets. These results demonstrate that Progressive Thought Encoding not only improves reasoning accuracy but also makes RL training of LRMs substantially more efficient and scalable under real-world memory constraints.

## 1 Introduction

Large reasoning models (LRMs) (Plaat et al., 2024; Li et al., 2025b; Huang and Chang, 2022) are emerging as a new paradigm that extends large language models (LLMs) with enhanced capacity for multi-step reasoning (Fu et al., 2023), symbolic manipulation (Dave et al., 2024), and problem solving in real-world scenarios (Xu et al., 2024). Unlike conventional LLMs that rely primarily on scale and corpus size for improved performance, LRMs explicitly emphasize reasoning-oriented training signals and architectural design, making them particularly well suited for domains such as mathematics (Shao et al., 2024), science (Schmidgall et al., 2025), and programming (Wang et al., 2024). As these models continue to achieve impressive results on increasingly complex benchmarks (Phan et al., 2025; Wang et al., 2023), the focus of research has gradually shifted from pursuing raw capabilities to improving efficiency in training and deployment (Wu et al., 2025; Feng et al., 2025).

Reinforcement learning (RL) (Kaelbling et al., 1996) has become the standard approach for aligning and improving large reasoning models (LRMs) during post-training, with methods such as PPO (Schulman et al., 2017), GRPO (Guo et al., 2025), and related algorithms (Zheng et al., 2025a; Yu et al., 2025; Li et al., 2025a) providing fine-grained control over reasoning behavior. However, RL suffers from a fundamental efficiency bottleneck: outcome-based rewards are sparse and only available after completing long sequences of actions (Yang et al., 2025), during which autoregressive decoding dominates memory and compute resources. The length of these trajectories, or chain-of-thought (CoT) reasoning, scales with task complexity, yielding longer rollouts for more challenging problems. Such extended CoT sequences significantly increase post-training and inference costs.

A natural strategy to address this challenge is to bound memory through sliding-window caches (Alizadeh et al., 2024) or dynamic pruning of past tokens (Fu et al., 2024). However, these approaches

---

[*]Work done during the internship at Microsoft Research. [†]Correspondence to: *zeliang.zhang@rochester.edu*, *xiaodl@microsoft.com*.

often degrade reasoning quality, as discarding intermediate thoughts weakens the model's ability to integrate long-range context (Cai et al., 2024). This degradation not only impacts reasoning accuracy at inference time but also reduces sample quality during the rollout stage, thereby hindering the effectiveness of training. This tension raises a critical question: *can LRMs be trained efficiently under strict memory budgets without sacrificing reasoning accuracy?*

In this work, we introduce **Progressive Thought Encoding**, a parameter-efficient fine-tuning method designed to address this bottleneck. Rather than discarding evicted tokens, our approach encodes their information into fixed-size vector representations that preserve long-context understanding under limited caches. We dynamically embed this contextual information into lightweight LoRA adapters, allowing the model to retain key reasoning signals without increasing cache size. By integrating this online adaptation into reinforcement learning, our method reduces peak memory usage during post-training. The learned adapters further enable the model to maintain strong reasoning performance under constrained computational budgets during inference.

We evaluated our method on three representative models: `Qwen2.5-4B-Instruct`, `Qwen2.5-7B-Instruct`, and `DeepSeek-R1-Distill-Llama-8B`, across six challenging mathematical reasoning benchmarks. Our approach consistently outperforms vanilla RL training, achieving up to a 23.4% improvement in reasoning accuracy on AIME while reducing GPU memory usage by nearly 50%. These results demonstrate that cache-aware reinforcement learning not only makes training large reasoning models more efficient but also improves their reasoning capabilities.

Our contributions can be summarized as follows:

- We identify the fundamental inefficiency of RL training for LRMs under long rollouts and formalize it as a cache-constrained optimization problem.

- We propose Progressive Thought Encoding, a parameter-efficient fine-tuning technique that learns from evicted tokens to preserve reasoning capacity under bounded memory.

- Through extensive experiments on open-weight models and math benchmarks, we show that our method substantially improves both training efficiency and inference robustness, setting a new standard for scalable reasoning model training.

## 2 RELATED WORK

**Test-time Learning of LLMs**. Test-time learning (TTL) explores how LLMs can adapt to new tasks or distributions without offline retraining (Hu et al., 2025). The most basic form is in-context learning (Dong et al., 2022), where demonstrations embedded within the prompt elicit task-specific behavior, while retrieval-augmented generation (RAG) extends this idea by providing task-relevant documents at inference (Gao et al., 2023; Han et al., 2024; Cheng et al., 2025). More advanced methods allocate additional computation for reasoning during inference, including tree-of-thought search (Yao et al., 2023), self-consistency across multiple reasoning paths (Wang et al., 2022), and iterative refinement (Madaan et al., 2023). Another line of work investigates gradient-based updates at test time, such as test-time training (Zuo et al., 2025) and entropy minimization techniques (Zhang et al., 2025b; Agarwal et al., 2025), while recent theory establishes connections between instruction tuning–based TTL and low-rank parameter updates in LLMs (Dherin et al., 2025).

**Parameter-efficient Fine-tuning of LLMs**. Since the introduction of Low-rank Adaptation (LoRA) (Hu et al., 2022), numerous parameter-efficient fine-tuning (PEFT) methods have been developed to improve the efficiency of adapting large language models (LLMs) to downstream tasks, including QLoRA (Dettmers et al., 2023), LiSA (Pan et al., 2024), and prefix-tuning (Li and Liang, 2021). While these approaches primarily focus on offline task adaptation, recent work has extended low-rank techniques to enable dynamic test-time learning, such as generative adapters (Chen et al., 2024) and stream adapters (Muhtar et al., 2024), which allow LLMs to adapt on-the-fly to new inputs or distributional shifts, thus enhancing robustness and flexibility.

## 3 METHODOLOGY

### 3.1 NOTATION AND PRELIMINARIES

**Attention and the KV cache as memory**. In the prefilling stage, given a sequence $(x_1, \ldots, x_t)$, each token $x_i$ is mapped to a hidden state $h_i$, which is then projected into query, key, and value vectors, *i.e.*, $q_i = W_Q h_i, k_i = W_K h_i, v_i = W_V h_i$, where $W_Q$, $W_K$, and $W_V$ are learnable weight matrices.

Let $K_t = [k_1, \ldots, k_t]$ and $V_t = [v_1, \ldots, v_t]$ denote the cache of keys and values up to step $t$. The attention output for token $x_t$ is given by

$$o_t = \mathrm{softmax}\left(\frac{q_t K_t^\top}{\sqrt{d}}\right) V_t.$$

During the decoding stage, for the next token $x_{t+1}$, we first compute its query $q_{t+1}$, and then let it attend over the extended KV cache:

$$o_{t+1} = \mathrm{softmax}\left(\frac{q_{t+1}[K_t, k_{t+1}]^\top}{\sqrt{d}}\right) [V_t, v_{t+1}].$$

Thus, the KV cache grows incrementally with each new token, serving as the memory that avoids redundant computation during autoregressive decoding and improves long-context understanding.

**GRPO for Reinforcement Learning in LLMs**. Grouped Reinforcement Policy Optimization (GRPO) is a policy gradient method designed to fine-tune large language models. Unlike classical RLHF approaches, GRPO discards the need for a critic model and instead samples multiple candidate completions per prompt, groups them, and assigns rewards at the group level.

Given a prompt $p$, the model generates $n$ completions $\{y_1, \ldots, y_n\}$ at the rollout stage. Then, each completion $y_i$ is assigned a raw score $s_i$ by a reward model, which is then normalized within the group to produce variance-reduced rewards:

$$r_i = \frac{s_i - \frac{1}{n}\sum_{j=1}^{n} s_j}{\sqrt{\frac{1}{n}\sum_{j=1}^{n}(s_j - \bar{s})^2 + \epsilon}}, \quad \bar{s} = \frac{1}{n}\sum_{j=1}^{n} s_j.$$

The policy is updated to maximize the expected reward while staying close to a reference policy $\pi_{\mathrm{ref}}$:

$$\mathcal{L}_{\mathrm{GRPO}}(\pi) = \mathbb{E}_{y \sim \pi(\cdot|p)}\Big[r(y) - \beta\,\mathrm{KL}\big(\pi(\cdot|p)\,\|\,\pi_{\mathrm{ref}}(\cdot|p)\big)\Big], \tag{1}$$

where $r(y)$ is the group-normalized reward and $\beta$ controls the KL regularization strength. Using relative rewards within each group, GRPO provides stable training signals without a critic and aligns naturally with autoregressive generation in LLMs.

### 3.2 CHALLENGES FOR EFFICIENT RL TRAINING

Difficult tasks often require long reasoning trajectories (Yang et al., 2025), *i.e.*, generating more tokens to obtain high-quality solutions for reward computation. The effectiveness of passive test-time scaling (Muennighoff et al., 2025) further underscores the importance of extended reasoning in solving difficult problems. However, this demand for longer generations directly amplifies the inefficiency of the rollout stage, which has been identified as the primary bottleneck to RL training (Zheng et al., 2025b; Han et al., 2025; Zhang et al.; 2025a). Despite the use of KV caching to avoid redundant computation, rollouts still dominate both time and memory costs due to continuous autoregressive decoding, making efficient training particularly challenging under outcome-based reward settings.

A natural approach to mitigating memory consumption is to adopt a dynamic sliding window strategy for the KV cache (Zhang et al., 2023), thus keeping memory usage approximately constant even as the roll-out sequences grow longer. However, aggressive token drop can significantly impair long-sequence understanding and generation (Jin et al., 2024; Cai et al., 2024), which in turn weakens the model's reasoning ability during rollouts and ultimately reduces training effectiveness. As illustrated in table 1, applying a sliding-window cache to RL training of Qwen models leads to a clear performance drop compared to training with the full cache of all tokens. This naturally raises a critical question: *can we maintain a constant-capacity cache window while still enabling the reasoning model to effectively "see" all previous tokens for efficient reasoning?*

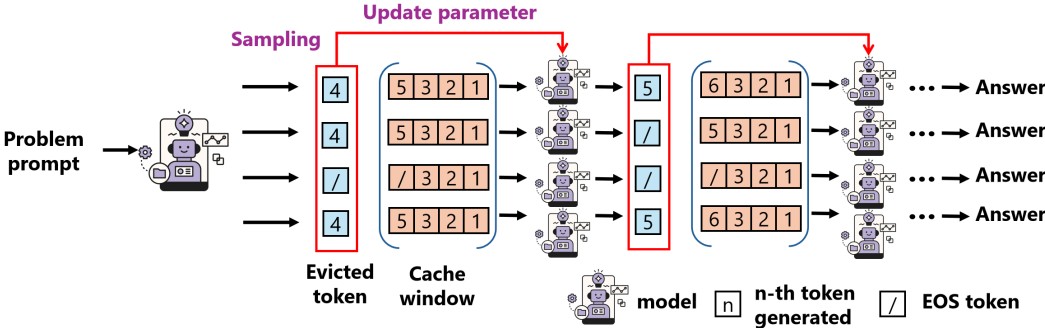

Figure 1: Overview of our method. During the rollout process, the model continuously learns the dropped tokens to achieve a balance between generation efficiency and long-term memory.

To formalize this challenge, we modify the standard GRPO formulation by redefining the rollout distribution. In the original objective, a trajectory $y$ is sampled under the full-cache policy $\pi_\theta(\cdot \mid p)$. In our setting, the trajectory is instead generated under a cache policy $D$, which prunes the KV cache online during decoding. At each step $t$, $D$ selects a pruned context $\mathcal{C}_t^D = \mathrm{CachePrune}_D(p, y_{<t})$, and the token distribution becomes

$$\pi_\theta^D(y \mid p) = \prod_{t=1}^{T} \pi_\theta\big(y_t \mid \mathcal{C}_t^D\big). \tag{2}$$

Accordingly, the cache-aware GRPO objective is

$$\mathcal{L}_{\mathrm{GRPO}}^D(\theta_g; \theta_{\mathrm{ref}}) \;=\; \mathbb{E}_{y \sim \pi_{\theta_g}^D(\cdot \mid p)}\Big[r(y) \;-\; \beta\,\mathrm{KL}\big(\pi_{\theta_g}^D(\cdot \mid p) \,\big\|\, \pi_{\theta_{\mathrm{ref}}}(\cdot \mid p)\big)\Big], \tag{3}$$

where $\theta_g$ denotes the parameters of the generating model under partial-cache rollouts, and $\theta_{\mathrm{ref}}$ is a reference model that operate with the full cache. Given a task prompt after the model training, we expect $\pi_{\theta_g^*}(y \mid p) \approx \pi_{\theta^*}(y \mid p)$, where $\theta_g^*$ and $\theta^*$ are optimized from eq. (3) and eq. (1) respectively.

### 3.3 OUR APPROACH: LEARNING THINK TOKENS PRIOR TO EVICTION

Motivated by prior work on dynamically adapting models to novel inputs at test time (Chen et al., 2024; Muhtar et al., 2024), we take a different approach from simply discarding the evicted thinking tokens. Instead, we first learn from these tokens to update a small set of parameters $\theta_g$, enabling the test-time policy $\pi^D\theta_g(y \mid p)$ to approximate the full-cache policy $\pi\theta(y \mid p)$ under a given eviction strategy $D$.

Specifically, for a given question $x$, during the rollout stage, we continuously decode next thinking tokens $\{y_1, \ldots, y_l\}$ based on the policy $\pi_\theta^D(y \mid p)$ until the KV cache is full. Based on the token eviction strategy $D$, earlier tokens $\{y_{e_1}, \ldots, y_{e_m}\}$ will be evicted from the cache. Rather than discarding these tokens, we use them to update a compact latent representation with the help of global query vector $q_g$, which serves as a learnable summary of all evicted context

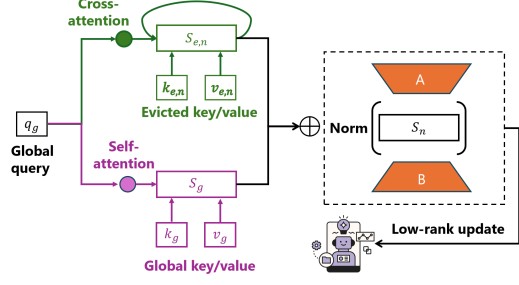

Figure 2: The computation of context state $S$.

encountered so far. The update to the LoRA weights is computed as

$$\triangle W = A \underbrace{\Big(\big((W_Q^a q_g)(W_K^a K_e)^T\big)(W_K^a V_e)\Big)}_{S_e} B, \tag{4}$$

where we denote $q_g$ as global latent query that aggregates information from evicted tokens, $W_Q^a, W_Q^a$ and $W_V^a$ as the weight matrices to map the global query tokens $q_g$, evicted key $K_e$ and value tokens

$V_e$ into the compressed latent space, $A$ and $B$ as the weight matrices to map the evicted context state $S_e$ computed by the evicted tokens to the model weights.

The model then continues decoding $\{y_{l+1}, \dots\}$ under the updated policy $\pi_{\theta'}^D(y \mid p)$, where $\theta' = \theta + \triangle W$. Each time the cache fills, we compute a new evicted context state $S_e'$ and update $S_e \leftarrow Normalize(S_e + S_e')$, and recompute $\triangle W$ accordingly.

To bootstrap adaptation, before processing any evicted tokens we initialize the context state with learnable global tokens as $S_e = \left(W_Q^a q_g \left(W_K^a k_g\right)^\top\right) W_V^a v_g$, where we define $h_g$ as the global tokens and $q_g = W_Q h_g$, $k_g = W_K h_g$, and $v_g = W_V h_g$. This initialization makes $q_g$ an explicit carrier of evicted-context information, enabling streaming adaptation while keeping memory usage constant. The full computation is illustrated in fig. 2.

**The selection of $D$ during training**. In our training setup, all question tokens are permanently retained in the cache, while a simple sliding-window eviction strategy is applied only to the thinking tokens. This straightforward design supports efficient batch operations across samples, whereas more sophisticated importance-based eviction would incur additional computational overhead. The decision to always keep question tokens is directly motivated by the sink-token mechanism in (Zhang et al., 2023), as both serve to anchor and preserve the prompt context, ensuring that the model maintains stable grounding even when the chain-of-thought becomes very long.

## 4 EVALUATIONS

Table 1: Comparison of methods across different models on benchmark datasets. The best average performance per model is highlighted in bold. *Note: Benchmark improvements are reported relative to Baseline, while FLOPs/Memory reductions are reported relative to LoRA.*

| Studied Models | Methods | Maximum TFLOPs of Attention | Peak GPU Mem. (%) | Mean GPU Mem. (%) | Math500 pass@1 | Olympiad pass@1 | Minerva Math pass@1 | AMC pass@1 | AIME2024 pass@16 | AIME2025 pass@16 | Avg. |
|---|---|---|---|---|---|---|---|---|---|---|---|
| Qwen2.5-3B-Instruct | Baseline | – | – | – | 50.8 | 27.2 | 16.1 | 34.3 | **20.0** | 13.3 | 26.9 |
| | LoRA | 4.2 | 82.8 | 63.5 | 53.2$_{+2.4}$ | 27.8$_{+0.6}$ | 15.9$_{-0.2}$ | 35.9$_{+1.6}$ | 20.0$_{0.0}$ | 16.7$_{+3.4}$ | 28.2$_{+1.3}$ |
| | LoRA$_c$ | 2.6$_{-1.6}$ | 38.0$_{-44.8}$ | 31.0$_{-32.5}$ | 50.0$_{-0.8}$ | 27.7$_{+0.5}$ | 16.1$_{0.0}$ | 33.1$_{-1.2}$ | 16.7$_{-3.3}$ | 10.0$_{-3.3}$ | 25.6$_{-1.3}$ |
| | Ours | 2.7$_{-1.5}$ | 45.3$_{-37.5}$ | 32.6$_{-30.9}$ | **54.0**$_{+3.2}$ | **29.0**$_{+1.8}$ | **16.2**$_{+0.1}$ | **45.0**$_{+10.7}$ | 20.0$_{0.0}$ | 16.7$_{+3.4}$ | **30.1**$_{+3.2}$ |
| Qwen2.5-7B-Instruct | Baseline | – | – | – | 56.8 | 34.7 | 18.5 | 48.4 | 23.3 | 16.6 | 33.1 |
| | LoRA | 5.7 | 85.8 | 59.3 | 59.4$_{+2.6}$ | **38.7**$_{+4.0}$ | 23.4$_{+4.9}$ | 50.6$_{+2.2}$ | 30.0$_{+6.7}$ | 26.7$_{+10.1}$ | 38.1$_{+5.0}$ |
| | LoRA$_c$ | 3.5$_{-2.2}$ | 63.1$_{-22.7}$ | 45.4$_{-13.9}$ | **61.2**$_{+4.4}$ | 35.9$_{+1.2}$ | 23.7$_{+5.2}$ | **52.5**$_{+4.1}$ | 20.0$_{-3.3}$ | 26.7$_{+10.1}$ | 36.7$_{+3.6}$ |
| | Ours | 3.6$_{-2.1}$ | 67.2$_{-18.6}$ | 48.6$_{-10.7}$ | **61.2**$_{+4.4}$ | **38.7**$_{+4.0}$ | 25.3$_{+6.8}$ | **52.5**$_{+4.1}$ | 30.0$_{+6.7}$ | **30.0**$_{+13.4}$ | **39.6**$_{+6.5}$ |
| DeepSeek-R1-Distill-Llama-8B | Baseline | – | – | – | 53.6 | 28.7 | 15.6 | 42.5 | 20.0 | 20.0 | 30.1 |
| | LoRA | 7.4 | 88.7 | 53.5 | 57.4$_{+3.8}$ | 35.3$_{+6.6}$ | 18.3$_{+2.7}$ | 55.0$_{+12.5}$ | 23.3$_{+3.3}$ | 20.0$_{0.0}$ | 34.9$_{+4.8}$ |
| | LoRA$_c$ | 4.6$_{-2.8}$ | 59.1$_{-29.6}$ | 47.1$_{-6.4}$ | 54.2$_{+0.6}$ | 31.9$_{+3.2}$ | 16.0$_{+0.4}$ | 45.0$_{+2.5}$ | 36.7$_{+16.7}$ | 26.7$_{+6.7}$ | 35.1$_{+5.0}$ |
| | Ours | 4.6$_{-2.8}$ | 59.8$_{-28.9}$ | 46.8$_{-6.7}$ | **57.6**$_{+4.0}$ | **39.7**$_{+11.0}$ | 16.5$_{+0.9}$ | **60.0**$_{+17.5}$ | **56.7**$_{+36.7}$ | **43.3**$_{+23.3}$ | **45.6**$_{+15.5}$ |

### 4.1 EXPERIMENTAL SETUP

**Models.** We evaluate our method on three representative open-weight instruction-tuned models of varying scales and architectures: (1) `Qwen2.5-3B-Instruct` (Team, 2024), a 4.1B-parameter transformer with 32 decoder layers, a hidden dimension of 4,096, 32 attention heads (128 dimensions per head), and rotary positional encodings; (2) `Qwen2.5-7B-Instruct` (Team, 2024), a mid-scale 7.2B-parameter model with 32 decoder layers, hidden size of 5,120, and 40 attention heads. Its architecture follows the same design principles as the 4B variant but with larger hidden width and attention capacity; (3) `DeepSeek-R1-Distill-Llama-8B` (Guo et al., 2025; Vavekanand and Sam, 2024), an 8.0B-parameter model distilled from DeepSeek-R1 into LLaMA-3.1-8B. It comprises 32 transformer layers with hidden dimension 4,096, 32 attention heads, SwiGLU activation, and rotary embeddings. Compared with the original `LLaMA-3.1-8B` model, it has better capacity on long-sequence generation.

**Benchmarks and Metrics.** We conduct evaluations on six math reasoning benchmarks covering diverse difficulty levels and reasoning depth: (1) **Math500** (Hendrycks et al., 2021), a curated set of 500 challenging word problems requiring symbolic and multi-step reasoning; (2) **Olympiad-Bench** (He et al., 2024), 674 olympiad-style problems designed to test deep mathematical reasoning; (3) **Minerva Math** (Lewkowycz et al., 2022), 672 problems sampled from arXiv and textbooks, emphasizing symbolic manipulation; (4) **AMC** (American Mathematics Competitions, 2023), 40

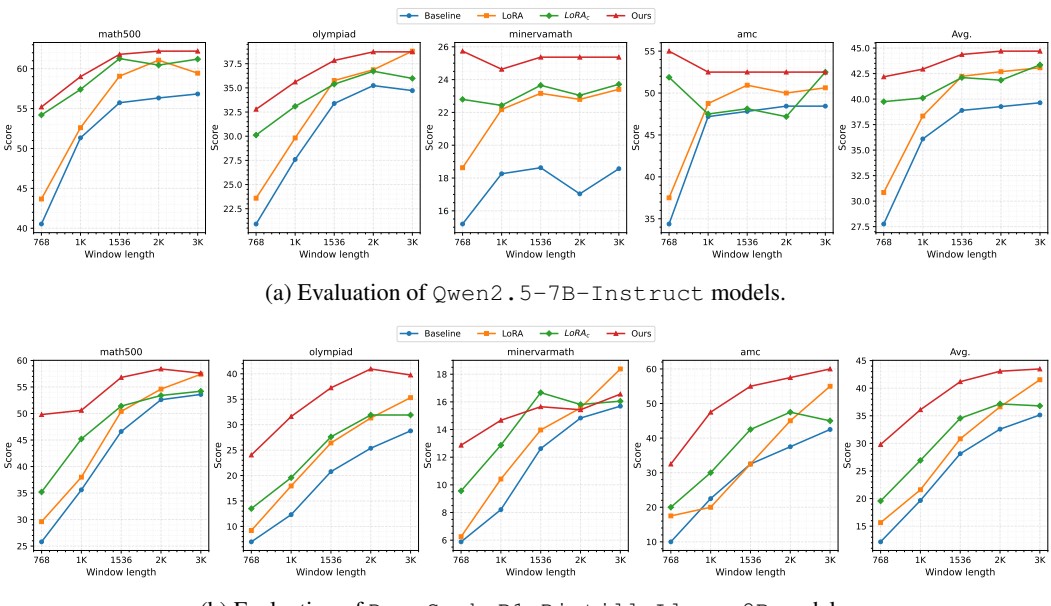

(a) Evaluation of `Qwen2.5-7B-Instruct` models.

(b) Evaluation of `DeepSeek-R1-Distill-Llama-8B` models.

Figure 3: Evaluation of `Qwen-7B-Instruct` and `DeepSeek-R1-Distill-Llama-8B` models trained by different methods on four benchmarks. We set the same maximum number of tokens for generation as 3072, and vary the KV cache window length from 768 to 3072. Each value corresponds to the mean pass@1 score over five independent runs.

middle- to high-school competition problems focused on combinatorics, number theory, and algebra; (5) **AIME2024** and **AIME2025** (Codeforces), recent American Invitational Mathematics Examination sets, each containing 30 highly challenging problems. Due to their extreme difficulty, AIME datasets are evaluated using the *pass@16* metric. For all other datasets, we report *pass@1*, averaged over 5 independent runs, to ensure fair and robust comparisons.

**Compared Methods.** We compare four approaches: (1) **Baseline**, the original model prior to RL training; (2) **LoRA**, RL-trained models with low-rank adaptation applied; (3) **LoRA$_c$**, RL-trained models with LoRA and a sliding-window cache for token eviction; (4) **Ours**, RL-trained models using our proposed method, where evicted tokens are explicitly learned before being discarded.

**Implementation Details.** Unless otherwise specified, the maximum sequence length during rollout is set to 3072, with a global batch size of 512. We use the DAPO-Math-17K dataset (Yu et al., 2025) as our training dataset. We use the Adam optimizer with a learning rate of $1 \times 10^{-5}$ and a maximum gradient norm of 1.0. The rank of LoRA and our method is fixed at 32. For LoRA$_c$ and our method, the sliding-window cache size is set to the maximum question length in the current micro-batch, with 25% of tokens evicted upon cache saturation to improve efficiency during training and inference. Our method additionally employs 32 global tokens. All models are trained until convergence, and experiments are conducted on 8 NVIDIA A100 GPUs (40 GB each).

### 4.2 EVALUATION ON MATH REASONING TASKS

We first evaluate the training efficiency and task performance of the trained models using different methods. Training efficiency is quantified using three metrics: (i) maximum TFLOPs required by attention, (ii) peak GPU memory utilization, and (iii) mean GPU memory utilization across training. These jointly reflect the computational and memory efficiency of the different cache strategies. Table 1 reports the results.

`Qwen2.5-3B-Instruct`. Full-cache LoRA attains 28.2% average accuracy but requires 4.2 TFLOPs and nearly 83% peak memory usage. LoRA$_c$ reduces peak memory to 38% but accuracy drops to 25.6%. In contrast, the proposed method achieves **30.1%**, the highest across all methods, while requiring only 2.7 TFLOPs and 45% peak memory. This demonstrates that naive eviction

Table 2: AIME2024 and AIME2025 pass@16 results (%). Maximum generation length is 6,144 tokens. KV cache window sizes range from 768 to 1,536. *Note: Improvements are reported relative to Baseline.*

| Dataset | Method | Qwen2.5-4B-Instruct | | | | Qwen2.5-7B-Instruct | | | | DeepSeek-R1-Distill-Llama-8B | | | |
|---|---|---|---|---|---|---|---|---|---|---|---|---|---|
| | | 768 | 1024 | 1536 | Avg. | 768 | 1024 | 1536 | Avg. | 768 | 1024 | 1536 | Avg. |
| AIME2024 | Baseline | 10.0 | 16.6 | 20.0 | 15.53 | 23.3 | 13.3 | 23.3 | 19.97 | 3.3 | 3.3 | 20.0 | 8.87 |
| | LoRA | $10.0_{0.0}$ | $13.3_{-3.3}$ | $20.0_{0.0}$ | $14.43_{-1.1}$ | $10.0_{-13.3}$ | $23.3_{+10.0}$ | $30.0_{+6.7}$ | $21.10_{+1.1}$ | $3.3_{0.0}$ | $3.3_{0.0}$ | $23.3_{+3.3}$ | $9.97_{+1.1}$ |
| | $LoRA_c$ | $13.3_{+3.3}$ | $13.3_{-3.3}$ | $16.7_{-3.3}$ | $14.43_{-1.1}$ | $16.6_{-6.7}$ | $20.0_{+6.7}$ | $20.0_{-3.3}$ | $18.87_{-1.1}$ | $6.7_{+3.4}$ | $16.7_{+13.4}$ | $36.7_{+16.7}$ | $10.03_{+1.2}$ |
| | Ours | $16.7_{+6.7}$ | $20.0_{+3.4}$ | $20.0_{0.0}$ | $18.90_{+3.4}$ | $26.6_{+3.3}$ | $26.6_{+13.3}$ | $30.0_{+6.7}$ | $27.73_{+7.8}$ | $26.7_{+23.4}$ | $30.0_{+26.7}$ | $56.7_{+36.7}$ | $37.80_{+28.9}$ |
| AIME2025 | Baseline | 6.7 | 13.3 | 13.3 | 11.10 | 10.0 | 16.7 | 16.6 | 14.43 | 6.7 | 10.0 | 20.0 | 12.23 |
| | LoRA | $6.7_{0.0}$ | $6.7_{-6.6}$ | $16.7_{+3.4}$ | $10.03_{-1.1}$ | $6.7_{-3.3}$ | $23.3_{+6.6}$ | $26.7_{+10.1}$ | $18.90_{+4.5}$ | $6.7_{0.0}$ | $10.0_{0.0}$ | $20.0_{0.0}$ | $12.23_{0.0}$ |
| | $LoRA_c$ | $6.7_{0.0}$ | $10.0_{-3.3}$ | $10.0_{-3.3}$ | $8.90_{-2.2}$ | $20.0_{+10.0}$ | $26.7_{+10.0}$ | $26.7_{+10.1}$ | $24.47_{+10.0}$ | $6.7_{0.0}$ | $20.0_{+10.0}$ | $26.7_{+6.7}$ | $17.79_{+5.6}$ |
| | Ours | $10.0_{+3.3}$ | $16.7_{+3.4}$ | $16.7_{+3.4}$ | $14.47_{+3.4}$ | $23.3_{+13.3}$ | $26.7_{+10.0}$ | $26.7_{+10.1}$ | $25.60_{+11.2}$ | $26.7_{+20.0}$ | $30.0_{+20.0}$ | $43.3_{+23.3}$ | $33.34_{+21.1}$ |

severely harms reasoning performance, but eviction-aware training not only recovers but improves accuracy relative to full-cache LoRA.

`Qwen2.5-7B-Instruct`. The trade-off between accuracy and efficiency becomes more evident at larger scale. LoRA achieves 38.1% accuracy but incurs high memory cost (85.8% peak). $LoRA_c$ lowers memory to 63.1% but reduces accuracy to 36.7%. The proposed method achieves the best average accuracy (**39.6%**), while cutting FLOPs almost in half compared to LoRA (3.6 vs. 5.7). This suggests that eviction-aware training is particularly beneficial as model size increases.

`DeepSeek-R1-Distill-Llama-8B`. For the largest model, efficiency constraints dominate. Full-cache LoRA requires 7.4 TFLOPs and 89% peak memory. $LoRA_c$ reduces resource usage but sacrifices accuracy. By contrast, our method yields a marked performance gain, achieving **45.6%** average accuracy, a +10.7 improvement over LoRA, while consuming only 4.6 TFLOPs and 59.8% peak memory. The improvements are especially notable on challenging benchmarks such as AIME2024 (+33.4) and AIME2025 (+23.3).

## 4.3 Evaluation under Different Computational Budgets

To assess the robustness of different methods under constrained memory, we evaluate performance across progressively reduced KV cache sizes. In practice, such reductions correspond to tighter computational budgets during inference, where only a fraction of the activations can be stored.

Figure 3 summarizes results across multiple reasoning benchmarks, including Olympiad, MinervaMath, AMC, and Math500, where we set the maximum response length as 3,072. Each curve reports accuracy as the available cache decreases from full capacity to highly constrained settings. As expected, the Baseline and LoRA methods degrade rapidly with shrinking cache size, reflecting their reliance on complete historical context. $LoRA_c$ alleviates this issue to some extent by incorporating sliding-window adaptation learning from the training process, but its effectiveness remains limited when the window becomes narrow. In contrast, our method consistently sustains higher accuracy across all computational budgets, demonstrating resilience to cache truncation. Quantitatively, averaged across all datasets and cache settings, our approach achieves an accuracy of 39.37, compared to 32.99 for LoRA and 30.31 for the Baseline. This corresponds to relative improvements of +19.3% over LoRA and +29.9% over the Baseline. Importantly, these gains are achieved without requiring additional inference-time memory, as our method maintains constant cache usage regardless of the budget.

We further validate these findings on harder benchmarks, AIME2024 and AIME2025, which require longer chains of reasoning. Here, we allow up to 6,072 tokens for generation (exceeding the training setting) and set the maximum cache size to 1,536 tokens to accelerate decoding. We then report pass@16 scores across cache sizes $\{768, 1024, 1536\}$ in table 2. Across both years and all backbones, our method achieves the highest average performance. Relative to LoRA, the average gains are +6.63 / +6.70 on `Qwen2.5-7B-Instruct`, and +27.83 / +21.11 on `DeepSeek-R1-Distill-Llama-8B`. Improvements over the sliding-window Cache, *i.e.*, $LoRA_c$, are likewise substantial (e.g., +8.86 on `Qwen2.5-7B` AIME2024 and +27.77 on `DeepSeek-R1- Distill-Llama-8B` AIME2024), underscoring the limitations of naïve context truncation. More results on `Qwen-2.5-4B-Instruct` are provided in appendix B.

Table 3: Training efficiency comparison across different maximum generation lengths during rollout.

| Method | Vanilla | | With ours | | | |
|---|---|---|---|---|---|---|
| Generation Length | 3K | 6K | 3K | 4K | 5K | 6K |
| Peak GPU mem (%) | 88.7 | 95.6 | 59.8 | 60.2 | 60.1 | 60.4 |
| Mean GPU mem (%) | 53.5 | 64.3 | 46.8 | 46.9 | 46.7 | 47.6 |
| MATH-500 (pass@1) | 53.2 | 55.4 | 57.6 | 58.2 | 59.4 | 60.2 |

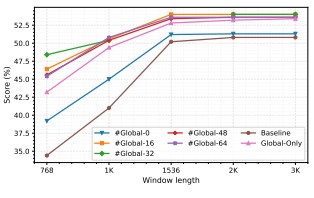

(c) Global tokens.

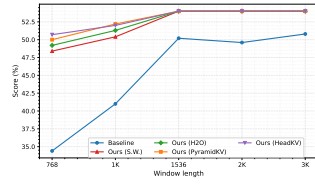

(d) Token dropping.

Figure 4: Ablation study on (a) global token usage and (b) token dropping strategies.

In summary, while our approach reduces training cost, particularly by lowering peak memory usage, without sacrificing task performance, these results further show that it also reduces inference cost by sustaining accuracy under tight cache budgets.

## 4.4 ABLATION STUDY AND DISCUSSION

**Progressive Thought Encoding Enables Scalable CoT RL Training**. We employ the proposed progressive encoding method to efficiently reduce memory consumption, particularly peak usage during training. By lowering memory requirements, we enable longer and more complex reasoning processes during the rollout stage. In this section, we present experiments demonstrating how the saved memory allows us to train `DeepSeek-R1-Distill-Llama` with larger maximum generation lengths, ranging from 4K to 6K tokens per rollout sample.

As shown in table 3, increasing the maximum generation length during rollout consistently improves reasoning performance on MATH-500. Meanwhile, progressive encoding keeps both peak and mean memory usage stable and significantly lower than vanilla RL training. Encouraging the model to generate longer outputs not only supports more extended reasoning but also leads to consistent gains on MATH-500. These results demonstrate that we can achieve longer reasoning with limited memory overhead, yielding better overall performance.

**Generation of long sequences for reasoning**. To assess the scalability of our method under extended reasoning trajectories, we evaluate the RL-trained DeepSeek-R1-Distill-Llama-8B model on MATH-500 using substantially longer generation lengths. During inference, we fix the context window to 1K tokens to impose a strict memory constraint, while varying the maximum generation length from 3K up to 64K tokens. This setup allows us to examine how well different approaches leverage increasingly long reasoning chains when the available KV cache remains limited The results are provided in fig. 5, showing that all methods benefit from longer reasoning sequences.

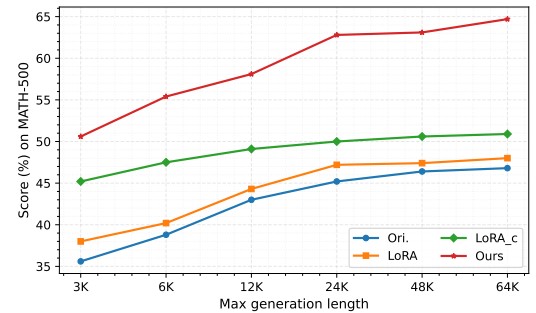

Figure 5: Performance on MATH-500 under a fixed 1K context window as the maximum generation length increases from 3K to 64K.

Across the entire length range, our method demonstrates the strongest scaling behavior. The original model, LoRA, and LoRA$_c$ show moderate improvements that gradually plateau as the sequence grows longer, whereas our approach continues to yield steady gains even at 64K tokens. This indicates that progressive thought encoding not only preserves reasoning information under tight cache budgets, but also scales favorably as reasoning trajectories extend far beyond the training rollout length.

**The use of global context tokens**. In our proposed method, we introduce *global tokens* to improve training efficiency. To evaluate their impact on model performance, we compare against several baselines: **(1) Baseline**, the original `Qwen-2.5-Instruct model`; **(2) Global-Only**, our method with the update of context state $S_e$ from evicted tokens disabled; **(3) #Global-0**, initializing $s_e$ with zero, effectively removing global token initialization; and **(4) #Global-16/32/48/64**, our method with the number of global tokens varied from 16 to 64. We conduct experiments on the MATH-500 dataset under different cache sizes $\{756, 1K, 1536, 2K, 3K\}$. The results are presented in fig. 3c.

It can be observed that disabling global tokens (#Global-0) yields only marginal improvements over the baseline. In contrast, integrating global tokens with the evicted token update of $S_e$ consistently enhances performance across different KV cache lengths, outperforming the Global-Only variant by a clear margin of $1.2\%$ at just 768 cached tokens. However, adding more global tokens does not always lead to better results: for example, #Global-64 underperforms compared with #Global-32 and #Global-16 at the most constrained cache length of 768 tokens.

**Integration with inference-time token dropping strategy**. In our work, we adopt the sliding window strategy for token eviction, which does not account for token importance. To address this limitation, we integrate several advanced token dropping strategies during generation and evaluate their performance on the MATH-500 dataset, including H2O (Zhang et al., 2023), PyramidKV (Cai et al., 2024), and HeadKV (Fu et al., 2024).

As shown in fig. 3d, compared to the sliding window eviction strategy, these advanced token dropping methods consistently improve reasoning performance, particularly under limited cache capacity. For example, with a cache window length of 768, the baseline model achieves a success rate of $34.4\%$. Using a sliding window cache increases performance to $48.4\%$, while HeadKV achieves the accuracy at $50.7\%$, narrowing the gap to full cache accuracy by $3.3\%$. These results demonstrate that token selection matters for reasoning efficiency.

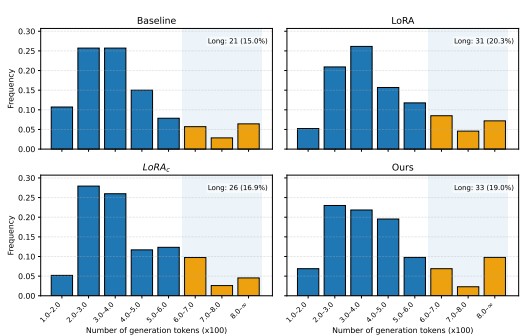

Figure 6: The statistics on the generation length.

However, these advanced strategies incur nontrivial cost. Integrating HeadKV during the rollout stage (batch size 512) increases iteration time from 19 to 26 minutes ($+37\%$ runtime) for a $+2.3\%$ accuracy gain.

Consequently, we retain the sliding-window approach for training and leave efficient integration of advanced token-dropping methods into RL rollouts as future work.

**On the length of generated response**. We also analyze the distribution of generated response lengths across different methods on the MATH-500 dataset. We set the maximum number of generation tokens to 3096, the cache window size to 768, and the number of sink tokens to 512, i.e., 256 tokens stored within the sliding window.

As shown in fig. 6, although $LoRA_c$ outperforms vanilla LoRA under a limited cache size (approximately $10\% \uparrow$, see fig. 3), most of these gains come from short responses, and only a few problems are solved with long responses. In contrast, our proposed method not only achieves the best overall reasoning performance under this setting but also maintains strong capability on long-form reasoning. These results support our claim that dynamically encoding evicted tokens into model weights enables the model to consistently "remember" them throughout the generation process.

*Why Progressive encoding can achieve better results*? Our method achieves higher accuracy because the progressive encoding of evicted tokens provides a continuous mechanism for preserving long-range reasoning information that would otherwise be lost under sliding-window truncation. Instead of discarding early thought tokens, their compressed contextual representations are folded into the LoRA weights, enabling the model to retain global reasoning signals even when only a small portion of the KV cache is visible. This acts as a form of denoising and incremental distillation, strengthening the model's ability to maintain coherent multi-step reasoning trajectories. Empirically, this leads to longer and more stable chains of thought during problem solving (see fig. 6), and substantially improves performance across constrained-cache settings. Together, these effects allow the model to approximate a full-context reasoner while operating under tight memory budgets, explaining the consistent gains over LoRA and sliding-window baselines.

## 5 CONCLUSION

We introduced Progressive Thought Encoding, a parameter-efficient fine-tuning approach that allows large reasoning models to train and infer effectively under limited computing resources. Rather than discarding evicted tokens, our method encodes their information into model weights, preserving long-context reasoning ability while substantially reducing memory and compute costs. Through experiments on three open-weight models and six challenging math reasoning benchmarks, we demonstrate consistent gains over LoRA and sliding-window cache baselines, achieving up to +23.4 absolute accuracy improvements on AIME2024/2025 while cutting peak memory nearly in half. Beyond boosting efficiency, our results show that cache-aware training enhances reasoning robustness under constrained computational budgets, enabling longer and more effective rollouts during RL training. We believe this work is a step toward scalable RL training for reasoning models and opens promising directions for adaptive eviction strategies, multimodal reasoning tasks, and integration with inference-time optimization techniques to further advance the efficiency–accuracy frontier.

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

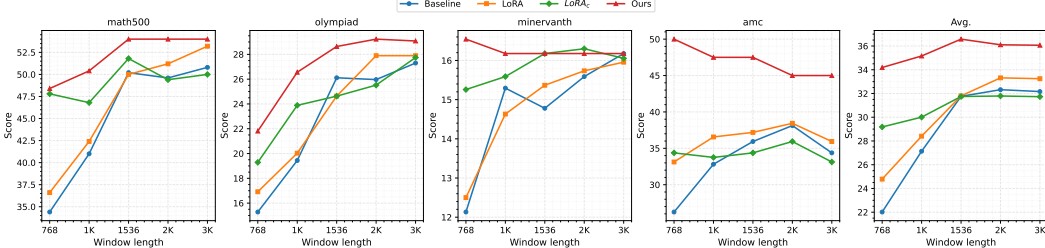

Figure A1: Evaluation of Qwen2.5-4B-Instruct models.

## A  THE USE OF LARGE LANGUAGE MODELS

In accordance with the ICLR 2026 policies on the use of Large Language Models (LLMs), we disclose that we used an LLM (OpenAI's ChatGPT) solely for writing assistance. Specifically, the model was employed to polish the language of the manuscript, including improving grammar, clarity, and readability.

No part of the model's output was used to generate research ideas, derive results, conduct experiments, or analyze data. All scientific contributions, including the design of experiments, implementation of methods, data analysis, and interpretation of results, are entirely the work of the listed authors, who take full responsibility for the content of this paper.

## B  RESULTS ON QWEN-2.5-4B-INSTRUCT

Following the settings in Section 4.3, we evaluate `Qwen-2.5-4B-Instruct` under different KV-cache budgets, with results shown in Figure A1. Across all four benchmarks (math500, olympiad, minervanth, and amc), our method (red curve) consistently outperforms the Baseline, LoRA, and LoRA variants. The gains are most pronounced at shorter window lengths (e.g., 768 and 1 K), where baseline models experience substantial accuracy degradation. For instance, on math500, our approach improves by more than 12 points over the baseline at 768 tokens, and it maintains its advantage even as the window length grows to 3 K. Similar trends appear on olympiad and amc, where our curve remains flat and robust while the baselines fluctuate or decline.

The rightmost panel shows the averaged results across all tasks, where our method consistently achieves the highest performance across the entire range of window lengths. Notably, our curve peaks around 1.5 K and remains stable thereafter, suggesting that our approach is not only more resilient to cache constraints but also scales gracefully with longer contexts. This demonstrates that training with cache-aware eviction leads to robust generalization and mitigates the performance drop observed in other fine-tuning strategies.

## C  IMPACT OF CACHE-EVICTION STRATEGY ON THE UPDATE OF $q_g$

To analyze how the cache-eviction strategy influences the update of the global latent vector $q_g$, we evaluate `Qwen-2.5-3B-Instruct` under different eviction ratios. The training setup matches that used in the main experiments, where a 25% ratio is applied during training. At inference time, however, we vary the ratio from 25% to 5% while keeping the context window fixed at 1024 tokens. The results are reported in Table A1.

Table A1: Effect of eviction ratios on MATH-500.

| Ratio | 25% | 20% | 15% | 10% | 5% |
|---|---|---|---|---|---|
| Score | 50.8 | 51.4 | 51.9 | 50.7 | 49.6 |

We observe that decreasing the eviction ratio initially improves performance: reducing the ratio to 15% yields the highest accuracy, suggesting that more frequent but smaller update steps enable $q_g$ to capture more fine-grained information from evicted tokens. However, when the ratio becomes too small (e.g., 5%), performance degrades noticeably. This indicates that overly fine-grained eviction

leads to noisier update signals with insufficient contextual content per step, resulting in unstable LoRA adaptation.

Overall, these results show that the eviction strategy plays a critical role in shaping the quality of the update signal for $q_g$. Moderate eviction ratios provide a more reliable balance between update frequency and information richness.

