# OpenReview forum: "Training Large Reasoning Models Efficiently via Progressive Thought Encoding"
_ICLR.cc/2026/Conference — ICLR 2026 Poster_

### Official Review · Reviewer_Dfk2 · 2025-10-31

**Soundness:** 2
**Presentation:** 2
**Contribution:** 2
**Rating:** 4
**Confidence:** 3

**Summary:**

This paper introduces the Progressive Thought Encoding method, a novel reasoning architecture for reasoning models. By encoding expelled intermediate reasoning tokens into fixed-size vectors and integrating them as a LoRA adapters, it avoids the high memory consumption of full-cache backtracking while preserving key reasoning signals. Experiments conducted on 3 open-source models and 6 mathematical reasoning benchmarks demonstrate that the proposed method achieves consistent improvements over LoRA and LoRA_c, while effectively reducing GPU peak memory usage and FLOPS.

**Strengths:**

1. The proposed method achieves better training results than PEFT methods on mainstream mathematical datasets.
2. Monitoring of FLOPs and GPU memory confirms that the approach of compressing tokens and reducing activations indeed contributes to efficient training.

**Weaknesses:**

1. Baselines. Although the ultimate goal is to reduce computational overhead, I am uncertain whether LoRA is an appropriate baseline. This work appears to be more aligned with studies on token compression and similar model architecture methods rather than PEFT methods. Have the authors considered comparing with relevant approaches? Please correct me if there is a misunderstanding.
2. Compatibility with existing infrastructure. The compression during reasoning seems to require per-sample maintenance of a series of related states, such as global query vectors. Can these modifications to the model architecture be easily implemented on transformers or vllm libraries to adapt to the existing infrastructure?
3. Context window. A 3K context window is generally insufficient for complex mathematical problems, especially for models trained with reasoning capabilities. The Distill-R1 model also has a longer output length. How does the performance change when the output length increases?

**Questions:**

Please consider discussing the issues raised in the weaknesses section.

4. I still have some confusion regarding the end-to-end procedure of RL training. For traditional RL, storing only the model's text responses is sufficient to compute the policy gradient. For the method proposed in this paper that adds LoRA heads to the model during test time, are any additional operations required?

---

> ### Author Response · Authors · 2025-11-24
>
> **W1:Baselines. Although the ultimate goal is to reduce computational overhead, I am uncertain whether LoRA is an appropriate baseline. This work appears to be more aligned with studies on token compression and similar model architecture methods rather than PEFT methods. Have the authors considered comparing with relevant approaches? Please correct me if there is a misunderstanding.**
> A:
> Thank you for the question. Our method is a *parameter-efficient fine-tuning (PEFT)* approach applied during RL training process which enable the test-time learning ability of LLMs, not a token-compression or architecture-level method. For this reason, **LoRA is the appropriate baseline**:
>
> 1. Our contribution modifies *how LoRA-style adapters learn under KV-cache limits*, so LoRA and LoRA_c isolate the effect we aim to study.
>
> 2. Token-compression methods (e.g., H2O, PyramidKV) target **inference-time cache reduction**, not **training-time adaptation**, and therefore address a different problem.
>
> 3. We still show compatibility with these token compression methods (H2O, HeadKV,PyramidKV, Fig. 4d in our paper), but they serve as complementary techniques to further improve the reasoning performance rather than baselines.
>
>
> **W2:Compatibility with existing infrastructure. The compression during reasoning seems to require per-sample maintenance of a series of related states, such as global query vectors. Can these modifications to the model architecture be easily implemented on transformers or vllm libraries to adapt to the existing infrastructure?**
> A: Yes. Our method is compatible with existing LLM infrastructures. All experiments were implemented using **TRL + HuggingFace Transformers** without requiring any intrusive architectural modifications. The additional per-sample states (e.g., global query vectors) are represented as lightweight learnable parameters, functionally similar to small linear layers, which can be efficiently broadcast across samples in standard batched computation. We have already implemented a proof-of-concept version in this framework, and the code will be released upon acceptance.
>
> Integrating PEFT-based dynamic updates into optimized inference engines such as vLLM and sglang presents additional challenges due to their tightly coupled and highly optimized KV-cache management mechanisms. In particular, recent attempts to support LoRA within the VERL + sglang backend still exhibit unresolved issues that impact stable LoRA integration during RL training.
> We have identified key bottlenecks in sglang and are actively working on extending our solution to vLLM. We are currently incorporating our approach into these platforms to enable more scalable RL training and to make Progressive Thought Encoding broadly accessible to the community.
>
>
>
>
>
> **W3:Context window. A 3K context window is generally insufficient for complex mathematical problems, especially for models trained with reasoning capabilities. The Distill-R1 model also has a longer output length. How does the performance change when the output length increases?**
> A: Thank you for the helpful suggestion. We agree that understanding performance under longer output lengths is important, especially for models designed for extended reasoning.
>
> To address this, we additionally evaluated our RL-trained DeepSeek-R1-Distill-Llama-8B model with **much longer generation lengths**, up to **64K tokens**, while keeping the **context window fixed to 1K** to test robustness under strict memory budgets. The results are shown below:
>
> | Method | 3K   | 6K   | 12K  | 24K  | 48K  | 64K  |
> |--------|------|------|------|------|------|------|
> | Ori.   | 35.6 | 38.8 | 43.0 | 45.2 | 46.4 | 46.8 |
> | LoRA   | 38.0 | 40.2 | 44.3 | 47.2 | 47.4 | 48.0 |
> | LoRA_c | 45.2 | 47.5 | 49.1 | 50.0 | 50.6 | 50.9 |
> | Ours   | 50.6 | 55.4 | 58.1 | 60.8 | 62.1 | 62.7 |
>
> For reference, we also ran LoRA under the **full-cache setting** using the same per-GPU batch size. However, full-cache inference runs into out-of-memory issues when the context exceeds 6K tokens. Under the maximum feasible lengths, LoRA obtains **56.8** at 3K and **58.3** at 6K. Our method, despite using only a **1K cache**, achieves **55.4** at 6K—only a **2.9-point drop** relative to full-cache LoRA—while remaining feasible at much longer lengths (up to 64K). It should be noted that our proposed approach can achieve the best result of around 63% with the same computation budget.
>
>
> Across all methods, performance increases as output length grows, but our approach shows the largest and most consistent improvements. Even with a short 1K context window, the model continues to benefit from longer reasoning traces, and our method scales more effectively than baseline LoRA and LoRAc.
>
> We will incorporate these results into the revised version and appreciate the reviewer’s feedback on evaluating longer-length reasoning.

---

> ### Author Response · Authors · 2025-11-24
>
> **Q1: I still have some confusion regarding the end-to-end procedure of RL training. For traditional RL, storing only the model's text responses is sufficient to compute the policy gradient. For the method proposed in this paper that adds LoRA heads to the model during test time, are any additional operations required?**
> A: Thank you for the question. Our method does **not** change the standard RL pipeline in terms of what must be stored or how the policy gradient is computed. The rollout data (prompt, generated tokens, and rewards) remain exactly the same as in traditional GRPO/PPO training.
>
> The only difference is internal to the **forward pass** during rollout:
>
> - When the KV cache evicts tokens, we update the LoRA parameters using the computed evicted-token state.
> - These updates are *lightweight, local, and do not require storing extra trajectories*.
> - The policy gradient is still computed on the final generated sequence exactly as in normal RL.
>
> In short, except the LoRA heads, the light weight weight matrix mapping tokens into LoRA weights and global tokens, there is **no additional storage or extra RL machinery is required**. The LoRA updates happen on-the-fly during generation, but the training loop, the logged data, and the gradient computation remain nearly the same as standard RL training. We report the FLOPS and GPU memory usage for the training in Tab. 1, where we also have low FLOPs and memory usage due to the token eviction compared with full cache LoRA training.

---

> ### Author Response · Authors · 2025-11-25
>
> Hi Reviewer Dfk2, we have attached our response here. Please let us know if you have any questions or concerns :) Thanks for all your efforts in reviewing our paper and giving us suggestions to improve it!!

---

### Official Review · Reviewer_MbcZ · 2025-11-01

**Soundness:** 2
**Presentation:** 3
**Contribution:** 3
**Rating:** 4
**Confidence:** 5

**Summary:**

Large scale chain of though RL runs with full cache make it memory and compute intensive. Simple solution is drop earlier context once cache limit hit or sliding window cache but authors show this comes at cost reasoning peformance. Further they propose to learn a global query vector $q_g$ from these evicted tokens KVs which can function like a memory. This $q_g$ is updated with a simple self-attention like update rule and then injected into a lightweight Lora Adapter via a small weight update $ \Delta W$. The model parameters now through these remember the evicted tokens. This adds minimal computational cost but allows scaling up RL by fixing cache size. They experiment with DAPO recipe on top of qwen-2.4 4/7B models and DeepSeek-R1-Distill-Llama-8B and show reasonable improvements over slidding window cache baseline.

**Strengths:**

- The idea of encoding the “to be evicted token” into latent vectors which are them embedded into LoRA adapter is very novel and clever. This makes the Lora adapter like a learned dynamic memory without blowing up KV cache linearly
- Well formulated and simple idea with clear experiments
- Good ablations and analysis

**Weaknesses:**

- Results and experimental setup is not convincing enough.
- Unclear if it will apply to full finetuned models compared to the LoRA only setting in this paper. Also unclear if this will translate over to larger models but I understand that will be outside the scope of this study.
- max sequence length of 3072 is not enough for reasoning RL runs and doesn’t give me confidence in the results especially given the authors tout this as benefitial for long reasoning RL. Only at longer context lengths the benefits or downside of this appraoch can be demonstrated.
- AIME25 still has variance at pass@16 so the slight improvment in those benchmarks is not conclusive
- Its also unclear why this approach should be benefitial over full-cache setup especially given the short max 6k context experiments. At larger contexts I can see how longer context can hurt sometimes. More thorough investigation is needed.
- $q_g$  is just one fixed vector for the whole evicted sequence. How much capacity can it even have? As we scale up the reasoning RL which can go upto 64k sequence length in frontier RL runs, I am not convinced this $q_g$  can hold that long context well. Ofc it can be better than no context of evicted tokens in sliding window cache but unclear how much we will lose compared to full cache.

**Questions:**

- What do the authors think is the impact of the cache eviction strategy on the q_g update would be. Eg bulk eviction or one at a time.
- Contribution [1] : Is that a novel contribution or already established in the community. Authors should clarify
- What do authors think about mamba hybrid models like nano-v2-12b https://huggingface.co/nvidia/NVIDIA-Nemotron-Nano-12B-v2 as they are optimized for inference. Its a hybrid model so cache still grows up but not to the same level. Also they will still able to model the long reasoning context much better.
- Comments on writing
    - In line 196 Table 1 is mentioned but we don’t know what lora_c is and its not explained so confusing to read
    - line 220 is not coherent
    - global query is not well introduced as being the latent that learns the evicted context.

---

> ### Author Response · Authors · 2025-11-24
>
> **W1:Results and experimental setup is not convincing enough.**
> A: Thank you for your suggestion. We have updated the experimental results in response to the concerns raised in the subsequent questions and revised the manuscript accordingly.
>
>
> **W2: Unclear if it will apply to full finetuned models compared to the LoRA only setting in this paper. Also unclear if this will translate over to larger models but I understand that will be outside the scope of this study.**
> A: Thank you for the question. As shown in Fig. 2 of our paper, our method is designed specifically around the *dynamic update of LoRA weights* during rollout using a sandwich-style structure. Because the mechanism operates through LoRA adapters, it does not directly target full-parameter fine-tuning.
>
> That said, we agree that understanding whether the method remains effective in the context of full fine-tuning is valuable. Following the same training setup, we conducted **full-parameter RL fine-tuning** experiments on Qwen2.5-3B-Instruct and Qwen2.5-7B-Instruct. The results are provided below.
>
> **Qwen2.5-3B-Instruct**
>
> | Method | Math500 | Olympiad | MinevaMath | AMC  | AIME24 | AIME25 | Avg  |
> |--------|---------|----------|------------|------|--------|--------|------|
> | Ori.   | 50.8    | 27.2     | 16.1       | 34.3 | 20.0   | 13.3   | 26.9 |
> | Full   | 51.3    | 28.5     | 16.2       | 34.8 | 20.0   | 13.3   | 27.4 |
> | LoRA   | 53.2    | 27.8     | 15.9       | 35.9 | 20.0   | 16.7   | 28.2 |
> | Ours   | 54.0    | 29.0     | 16.2       | 45.0 | 20.0   | 16.7   | 30.1 |
>
> **Qwen2.5-7B-Instruct**
>
> | Method | Math500 | Olympiad | MinevaMath | AMC  | AIME24 | AIME25 | Avg  |
> |--------|---------|----------|------------|------|--------|--------|------|
> | Ori.   | 56.8    | 34.7     | 18.5       | 48.4 | 23.3   | 16.6   | 33.1 |
> | Full   | 58.8    | 36.3     | 24.1       | 51.0 | 26.7   | 26.7   | 37.3 |
> | LoRA   | 59.4    | 38.7     | 23.4       | 50.6 | 30.0   | 26.7   | 38.1 |
> | Ours   | 61.2    | 38.7     | 25.3       | 52.5 | 30.0   | 30.0   | 39.7 |
>
> Across both models, **full-parameter tuning does not always outperform LoRA**, which is consistent with prior findings [a,b] in RL post-training settings. Importantly, our method achieves the best performance across all settings, including against full-parameter tuning, suggesting that dynamic LoRA updates remain effective even when compared with full-model RL optimization.
>
> [a] LoRA Without Regret - Thinking Machines Lab
> [b]Sidahmed, Hakim, et al. "Parameter efficient reinforcement learning from human feedback." arXiv preprint arXiv:2403.10704 (2024).

---

> ### Author Response · Authors · 2025-11-24
>
> **W3:max sequence length of 3072 is not enough for reasoning RL runs and doesn’t give me confidence in the results especially given the authors tout this as benefitial for long reasoning RL. Only at longer context lengths the benefits or downside of this appraoch can be demonstrated.**
> **W5: It's also unclear why this approach should be benefitial over full-cache setup especially given the short max 6k context experiments. At larger contexts I can see how longer context can hurt sometimes. More thorough investigation is needed.**
>
> **W6: $q_g$ is just one fixed vector for the whole evicted sequence. How much capacity can it even have? As we scale up the reasoning RL which can go upto 64k sequence length in frontier RL runs, I am not convinced this $q_g$
>  can hold that long context well. Ofc it can be better than no context of evicted tokens in sliding window cache but unclear how much we will lose compared to full cache.**
>
> Thank you for the insightful comments. To more directly assess the effectiveness of our method under long-sequence reasoning and to clarify its behavior relative to full-cache methods, we conducted additional experiments using significantly extended generation budgets. Specifically, we evaluated the RL-trained DeepSeek-R1-Distill-Llama-8B model on MATH-500 with **maximum generation lengths up to 64K tokens**, while fixing the **cache window to 1K** to impose strict memory constraints. The results are shown below:
>
> | Method | 3K   | 6K   | 12K  | 24K  | 48K  | 64K  |
> |--------|------|------|------|------|------|------|
> | Ori.   | 35.6 | 38.8 | 43.0 | 45.2 | 46.4 | 46.8 |
> | LoRA   | 38.0 | 40.2 | 44.3 | 47.2 | 47.4 | 48.0 |
> | LoRA_c | 45.2 | 47.5 | 49.1 | 50.0 | 50.6 | 50.9 |
> | Ours   | **50.6** | **55.4** | **58.1** | **62.8** | **63.1** | **64.7** |
>
> We additionally compared against LoRA under the **full-cache** setting using the same per-GPU batch size. Full-cache inference becomes infeasible beyond 6K due to OOM, but within the feasible range LoRA achieves **56.8** at 3K and **58.3** at 6K. Our approach, using only a **1K cache**, reaches **55.4** at 6K—just a **2.9-point drop**—while remaining viable up to 64K tokens and achieving the best result (64.7) under the same computation budget.
>
> These results directly address the reviewer’s concerns:
>
> 1. **On W3 (short max-length evaluation):**
>    All methods improve as generation length increases, but our approach shows the **largest and most sustained gains**, whereas the baselines plateau. This demonstrates that the proposed progressive thought encoding continues to extract useful reasoning signals even when sequences extend far beyond the training rollout length.
>
> 2. **On W5 (benefits relative to full-cache methods):**
>    Full-cache decoding rapidly reaches memory limits (already >95% peak usage at 6K during training). Our method is designed for realistic, resource-constrained RL settings where full-cache rollouts are impractical. Under the same compute budget, our approach not only remains feasible at long-sequence lengths but also achieves **the strongest overall performance**.
>
> 3. **On W6 (capacity of a single global latent \(q_g\)):**
>    We agree that a single global vector cannot perfectly approximate a full cache at extreme lengths. Our method deliberately trades representational fidelity for **constant memory usage**. The long-sequence results show that although \(q_g\) is a compressed summary, it still enables performance to **continue improving up to 64K tokens**, substantially narrowing the gap to full-cache decoding while maintaining scalability.
>
> We  incorporate these extended results and clarifications into the revised manuscript (updated in Fig. 5).
>
>
> **W4:AIME25 still has variance at pass@16 so the slight improvment in those benchmarks is not conclusive**
> A: Thank you for pointing this out. We agree that AIME25 can exhibit noticeable variance under the pass@16 metric. To mitigate this, in the main paper we already report **the mean pass@16 over 5 independent runs**, which is mentioned in the experiment setup.
>
> To further validate stability, we additionally report the **avg@16** metric (average accuracy across all 16 samples), which is less sensitive to variance. The results for Qwen2.5-7B-Instruct are:
>
>
>
>
> | Method | AIME2024 | AIME2025 | Avg  |
> |--------|----------|----------|------|
> | Ori.   | 6.2      | 4.1      | 5.2  |
> | LoRA   | 10.5     | 8.3      | 9.4  |
> | LoRA_c | 11.6     | 8.2      | 9.9  |
> | Ours   | 13.7     | 9.8      | 11.8 |
>
>
>
> These results show **consistent improvements** across both AIME24 and AIME25 under a lower-variance metric.
>
> Finally, we note that our core contribution is **improving RL training efficiency under strict memory constraints**. Even when improvements on certain benchmarks are modest, the method provides substantial reductions in compute and memory usage, which is a key focus of this work.

---

> ### Author Response · Authors · 2025-11-24
>
> **Q1:What do the authors think is the impact of the cache eviction strategy on the q_g update would be. Eg bulk eviction or one at a time.**
> A: Thank you for the question. The cache-eviction strategy indeed influences how the global vector  q_g  is updated.
>
> When fewer tokens are evicted at each step, q_g  receives **more frequent but smaller updates**, which can help it capture finer-grained information. However, if the eviction becomes too fine-grained, the updates can become noisy, leading to less stable learning.
>
> To study this effect, we ran experiments on Qwen2.5-3B-Instruct under the same training setup as in the paper. During training we used a 25% eviction ratio, but during inference we varied the ratio from 25% down to 5% with a fixed 1024-token window. The MATH-500 results are:
>
> | Eviction ratio | 25% | 20% | 15% | 10% | 5%  |
> |----------------|-----|-----|-----|-----|-----|
> | MATH-500       | 50.8| 51.4| 51.9| 50.7| 49.6|
>
> We observe that decreasing the eviction ratio initially **improves performance**, consistent with learning more fine-grained updates. But as the ratio becomes very small, performance **drops**, indicating noisy or unstable LoRA updates. This confirms that the eviction strategy directly affects the quality of the learned update signal for \( q_g \), and extremely small eviction steps are not beneficial.
>
> We add this analysis to the revised manuscript in Appendix C.
>
>
>
>
>
> **Q2:Contribution [1] : Is that a novel contribution or already established in the community. Authors should clarify**
> A: Our proposed approach is a novel contribution to efficient RL training under limited computational resources. Specifically, we introduce a mechanism that learns from the evicted tokens within the reasoning trajectory and integrates this information into the LoRA weights for dynamic updates, an aspect not explored in prior RL training work.
>
> Regarding the reference “[1]” mentioned in your comment, we are unsure which specific work it refers to and suspect it may have been omitted. We will revise the manuscript to clarify this point. Thank you for bringing it to our attention.
>
>
>
> **Q3:What do authors think about mamba hybrid models like nano-v2-12b https://huggingface.co/nvidia/NVIDIA-Nemotron-Nano-12B-v2 as they are optimized for inference. Its a hybrid model so cache still grows up but not to the same level. Also they will still able to model the long reasoning context much better.**
> A: Thank you for the thoughtful suggestion. Hybrid-state architectures such as Mamba-based or Nano-v2-12B models indeed offer notable advantages for inference efficiency by partially replacing transformer attention with state-space components, thereby slowing KV cache growth. However, these models still retain transformer layers whose KV cache continues to grow with sequence length, and therefore do not eliminate the memory pressure encountered during long autoregressive rollouts.
> More importantly, the vast majority of open large-scale reasoning models in production remain transformer-based, including GPT-oss, DeepSeek, and Qwen. As a result, KV cache growth and long-rollout memory bottlenecks remain the dominant practical challenge in current RL training pipelines.
> Crucially, our work targets a different dimension of the problem. We focus on preserving reasoning information when tokens must be evicted under strict KV memory budgets during long RL rollouts with outcome-based rewards. This challenge arises from prolonged token-by-token generation and the need to retain intermediate reasoning states, rather than from attention complexity alone.
> We therefore view hybrid-state models and Progressive Thought Encoding as complementary rather than competing. While hybrid architectures improve computational efficiency, our method improves the robustness of reasoning under cache truncation. Even in Mamba-hybrid settings, eviction remains necessary, and our approach provides a principled way to retain critical reasoning signals. We expect our method to remain beneficial when integrated with such architectures in future work

---

> ### Author Response · Authors · 2025-11-24
>
> **Q4: Comments on writing**
> A: Thanks for your comments! We have revised our manuscript.
>
> [1] LoRA_c: RL-trained models with LoRA and a sliding-window cache for token eviction;
> [2] Line 220 (now line 189)
> [3] update line 201 and 216–218 to give a more clear introduction of global query tokens.

---

> ### Author Response · Authors · 2025-11-25
>
> Hi Reviewer MbcZ, we have attached our response here. Please let us know if you have any questions or concerns :) Thanks for all your efforts in reviewing our paper and giving us suggestions to improve it!!

---

> ### Comment · Reviewer_MbcZ · 2025-11-26
>
> Thank you for your very strong and detailed rebuttal and running experimentation which alleviates quite a few of my concerns. I will raise my rating accordingly.

---

> > ### Author Response · Authors · 2025-11-26
> >
> > Thank you very much for raising score to 8! We really appreciate your suggestion and your time reviewing our paper:)

---

### Official Review · Reviewer_arKo · 2025-11-02

**Soundness:** 3
**Presentation:** 3
**Contribution:** 3
**Rating:** 6
**Confidence:** 4

**Summary:**

The paper proposes a parameter-efficient fine-tuning method under RL to address the compute resource challenges posed by long-context in difficult problems. The sliding-window caches or dynamic pruning of past tokens approaches, which affects inference accuracy. In contrast, the progressive thought encoding proposed in this paper, by updating the context state to preserve the complete context information as much as possible, ultimately improves the training effect while simultaneously reducing the computational cost.

**Strengths:**

1. The proposed method outperforms both LoRA and LoRA_c in terms of accuracy across various models and evaluation datasets, while requiring only a marginal increase in computational resources compared to LoRA_c.
2. The method's reliability is demonstrated through comprehensive evaluation across multiple benchmark datasets.

**Weaknesses:**

1. Among the three models evaluated in this paper, Qwen2.5-4B-Instruct and Qwen2.5-7B-Instruct are not LRM. Their output lengths are shorter compared to DeepSeek-R1-Distill-Llama-8B, making their persuasiveness less compelling.
2. The maximum sequence length is only 3072, lacking evaluation experiments at longer reasoning lengths.

**Questions:**

1. Experimental results for DeepSeek-R1-Distill-Qwen-7B are required;
2. Consider supplementing with evaluation experiments at longer inference lengths;
3. The analysis lacks an explanation for why this approach achieves higher accuracy compared to the LoRA.

---

> ### Author Response · Authors · 2025-11-24
>
> **W1&Q1: Among the three models evaluated in this paper, Qwen2.5-4B-Instruct and Qwen2.5-7B-Instruct are not LRM. Their output lengths are shorter compared to DeepSeek-R1-Distill-Llama-8B, making their persuasiveness less compelling.**
>
> **Experimental results for DeepSeek-R1-Distill-Qwen-7B are required;**
>
> A: We thank the reviewer for the constructive feedback. We agree that DeepSeek-R1-Distill-Llama-8B better reflects true LRM behavior due to its longer reasoning trajectories, and our primary conclusions on scalable long-chain reasoning are grounded in this model, where we observe the largest gains.
> The inclusion of Qwen2.5-4B-Instruct and Qwen2.5-7B-Instruct was intended to demonstrate generality across widely-used RL backbones, which are standard in prior reasoning-focused RL studies and develop strong reasoning capabilities after fine-tuning. They were not positioned as definitive LRMs, but as complementary evidence across varying reasoning depths.
> As you suggest, following the same setting in Tab 1 of our paper, we  conduct additional  experiments on DeepSeek-R1-Distill-Qwen-7B , providing a more representative evaluation under extended reasoning trajectories.
>
> **Results of DeepSeek-R1-Distill-Qwen-7B on MATH-500 under different cache sizes**
>
> | Method | 768  | 1024 | 1536 | 2048 | 3072 |
> |--------|------|------|------|------|------|
> | Ori.   | 18.6 | 28.0 | 36.8 | 46.0 | 56.8 |
> | LoRA   | 24.5 | 33.6 | 44.2 | 50.5 | 54.8 |
> | LoRA_c | 27.8 | 39.4 | 50.3 | 53.2 | 55.6 |
> | Ours   | **32.0** | **43.5** | **55.1** | **58.4** | **59.0** |
>
> These results show clear, consistent improvements from our approach across all window lengths, further supporting its effectiveness on both standard LLMs and reasoning-enhanced models.
>
>
>
>
>
> **W2&Q2: The maximum sequence length is only 3072, lacking evaluation experiments at longer reasoning lengths.**
>
>
> **Consider supplementing with evaluation experiments at longer inference lengths;**
>
>
> A: **[Why 3072]** Our paper focuses on training LLMs under limited computational resources, which constrains the maximum sequence length to 3K tokens during training (see Peak GPU memory). For Table 1 (excluding AIME), we therefore set the inference sequence length to 3K to match the training setup.
>
> **[Training and evaluation on 6K]** For AIME, which cannot be reliably solved within 3K tokens, we increase the maximum sequence length to 6K and report the results in Table 1 (AIME) and Table 2. In addition, Table 3 varies the rollout-token budget during RL training from 4K to 6K. These results indicate that even when trained with only 3K rollout tokens, the model is able to generalize its reasoning ability to significantly longer sequences (up to 2× in our experiments).
>
>
> **[Evaluation on longer generation sequence]** Following your suggestion, we further conduct experiments with longer inference lengths, up to 64K tokens. We evaluate our trained Deepseek-r1-Distill-Llamma-8B on the MATH-500 dataset, with settting its generation length from 3K to 64K and the context window as 1K to save memory and accelerate the generation.   The results are included below and have been incorporated into the revised manuscript.
>
>
> | Method | 3K   | 6K   | 12K  | 24K  | 48K  | 64K  |
> |--------|------|------|------|------|------|------|
> | Ori.   | 35.6 | 38.8 | 43.0 | 45.2 | 46.4 | 46.8 |
> | LoRA   | 38.0 | 40.2 | 44.3 | 47.2 | 47.4 | 48.0 |
> | LoRA_c | 45.2 | 47.5 | 49.1 | 50.0 | 50.6 | 50.9 |
> | Ours   | 50.6 | 55.4 | 58.1 | 60.8 | 62.1 | 62.7 |
>
>
> From this result, all methods improve with longer inference lengths, but our approach benefits the most. Ori. and LoRA show moderate gains, while LoRA_c scales better under the 1K context window. Our method achieves the largest and most consistent improvements, and its advantage grows as the generation length increases, demonstrating effective long-sequence reasoning even up to 64K tokens.
>
>
>
>
> **Q3: The analysis lacks an explanation for why this approach achieves higher accuracy compared to the LoRA.**
> A: Thank you for the insightful question.  Our method achieves higher accuracy because **the adaptive integration of evicted tokens into the LoRA weights** effectively acts as a **denoising** and **compression** mechanism.  Instead of discarding early reasoning tokens, the model absorbs their distilled information into a compact latent representation, allowing it to focus on more informative and globally relevant reasoning patterns.  As shown in our ablation study (Sec. 4.4), this process leads to longer and more coherent chains of thought during problem solving compared with standard LoRA, indicating stronger long-context reasoning ability.  We believe this enhanced ability to preserve and utilize long-range reasoning signals is a key factor behind the observed performance gains.  A more detailed discussion has been added to Sec. 4.4.

---

> ### Author Response · Authors · 2025-11-25
>
> Hi Reviewer arKo, we have attached our response here. Please let us know if you have any questions or concerns :) Thanks for all your efforts in reviewing our paper and giving us suggestions to improve it!!

---

### Official Review · Reviewer_J7E7 · 2025-11-03

**Soundness:** 2
**Presentation:** 3
**Contribution:** 2
**Rating:** 6
**Confidence:** 5

**Summary:**

The authors  introduced  a new memory efficient  RL method to post-train LLMs under under strict memory budgets. The key idea is to keep a  constant memory capacity cache window (similar to sliding window attention) . The authors also introduce global tokens . These global tokens are used with evicted from cache tokens  for weights update. The authors compared proposed method with 2 prior methods: (1) RL + LoRA (2) RL with LoRA and SWA.

**Strengths:**

Good practical method to reduce memory footprint and compute during RL stage for reasoning models

**Weaknesses:**

The comparison with prior work seems not complete.  The proposed method uses global tokens. Should not these tokens be also used in LoRA and LoRA_c for fair comparison?

I would recommend to remove sections 2.1,  2.2 and 2.3 from  the "related work" . This is common knowledge very weakly related to proposed method.

**Questions:**

The comparison with prior work seems not complete.  The proposed method uses global tokens. Should not these tokens be also used in LoRA and LoRA_c for fair comparison?

How do you explain that Mean GPU memory for "ours" is less than LoRA_c in the last row of Table 1?

---

> ### Author Response · Authors · 2025-11-24
>
> **W1&Q1: The comparison with prior work seems not complete. The proposed method uses global tokens. Should not these tokens be also used in LoRA and LoRA_c for fair comparison?**
> A: You are right that the use of global tokens should be considered in the baseline methods to properly isolate and evaluate the contributions of each component in our approach. In our ablation study (line 426, Sec. 4.4), we have already added global tokens to LoRA_c, referred to as Global-Only. Following your suggestion, we further incorporate global tokens into the standard LoRA setup, using the same configuration of Qwen-2.5-3B-Instruct training described in Sec. 4.4. The corresponding results are reported below and will been updated in the revised manuscript.
>
>
> | Method | Global Token | 3072 | 2048 | 1536 | 1024 | 768  |
> |--------|--------------|------|------|------|------|------|
> | LoRA   | None         | 53.2 | 51.2 | 50.0 | 42.4 | 36.6 |
> |        | Yes          | 53.6 | 51.9 | 50.4 | 44.5 | 38.3 |
> | LoRA_c | None         | 50.0 | 49.4 | 51.8 | 46.8 | 47.8 |
> |        | Yes          | 53.4 | 53.2 | 52.8 | 49.4 | 43.2 |
> | Ours   | Yes          | 54.0 | 54.0 | 54.0 | 52.6 | 49.5 |
>
>
>
>
> **W2: I would recommend to remove sections 2.1, 2.2 and 2.3 from the "related work" . This is common knowledge very weakly related to proposed method.**
> A: Thank you for the suggestion. We have removed Sections 2.1 and 2.2 from the related work, as they indeed cover general background that is not essential to our contribution. Since our method focuses on improving reasoning performance through a test-time learning mechanism—specifically, the dynamic update of LoRA weights—we retain Section 2.3, which is directly relevant to this aspect of our approach. The manuscript has been revised accordingly.
>
>
>
>
> **Q2: How do you explain that Mean GPU memory for "ours" is less than LoRA_c in the last row of Table 1?**
> A: DeepSeek-R1-Distill-Llama-8B tends to generate longer sequences compared with Qwen models, which increases GPU memory usage, as reflected in the peak GPU memory. As optimization progresses, the model learns to select better answers with fewer tokens, thereby reducing average GPU memory consumption. The lower memory usage of our method indicates that the model learns to solve the problem with a shorter reasoning path. In contrast, LoRA_c generates much longer sequences while achieving lower accuracy, suggesting weaker learning ability in this setting.
>
>
> We hope the clarifications and additional results fully address your concerns.

---

> ### Author Response · Authors · 2025-11-25
>
> Hi Reviewer J7E7, we have attached our response here. Please let us know if you have any questions or concerns :) Thanks for all your efforts in reviewing our paper and giving us suggestions to improve it!!

---

### Author Response · Authors · 2025-11-24

**To all reviewers:**
We sincerely thank all reviewers for the constructive and insightful feedback. We appreciate the consensus on the strengths of our work:
- **Novelty and practical relevance of the proposed idea** (J7E7, MbcZ, Dfk2), and
- **Comprehensive experiments demonstrating clear effectiveness across multiple benchmarks** (arKo, MbcZ, Dfk2).

In response to the concerns raised during the rebuttal period, we have conducted additional experiments, clarified technical details, and updated the manuscript accordingly. Key additions include:
- extended long-sequence evaluations up to **64K tokens**,
- new experiments on **DeepSeek-R1-Distill-Qwen-7B**,
- additional ablations on **global tokens**, **eviction strategies**, and **full-parameter tuning**, and
- clarifications on method design, memory efficiency, and applicability to existing infrastructures.

Updates of our manuscripts are highlighted in blue. We hope these updates fully address the reviewers’ questions and further strengthen the contribution of our paper. Thank you all:)


Main results are attached as follows,

Tab.1 Results of long-context reasoning of DeepSeek-R1-Distill-Llama-8B on Math-500. The size of KV cache is set as 1024.
| Method | 3K   | 6K   | 12K  | 24K  | 48K  | 64K  |
|--------|------|------|------|------|------|------|
| Ori.   | 35.6 | 38.8 | 43.0 | 45.2 | 46.4 | 46.8 |
| LoRA   | 38.0 | 40.2 | 44.3 | 47.2 | 47.4 | 48.0 |
| LoRA_c | 45.2 | 47.5 | 49.1 | 50.0 | 50.6 | 50.9 |
| Ours   | 50.6 | 55.4 | 58.1 | 60.8 | 62.1 | 62.7 |


Tab.2 Results of DeepSeek-R1-Distill-Qwen-7B on MATH-500 with different KV cache size
| Method | 768  | 1024 | 1536 | 2048 | 3072 |
|--------|------|------|------|------|------|
| Ori.   | 18.6 | 28.0 | 36.8 | 46.0 | 56.8 |
| LoRA   | 24.5 | 33.6 | 44.2 | 50.5 | 54.8 |
| LoRA_c | 27.8 | 39.4 | 50.3 | 53.2 | 55.6 |
| Ours   | **32.0** | **43.5** | **55.1** | **58.4** | **59.0** |

---

### Author Response · Authors · 2025-11-30

## **To the AC and All Reviewers**

We would like to once again thank the AC and the reviewers for their time, effort, and insightful feedback. Below is a consolidated summary of the contribution and concerns raised across the reviews, along with our corresponding responses.

In short, we have provided comprehensive and systematic responses to each of the reviewers’ comments. However, due to the shortened discussion period, we received a follow-up response from only one reviewer (MbcZ), and the average score increased from 5 to 6.

---

### **Main Contribution**

Our paper introduces a **new memory-efficient RL method** for post-training LLMs under **strict memory budgets**, achieving significant reductions in **GPU peak memory** and **FLOPs** (Reviewers **J7E7**, **arKo**, **MbcZ**, **Dfk2**).

We conducted **extensive experiments**, *three advanced models before rebuttal, plus one additional model during rebuttal*, across a broad suite of benchmarks to thoroughly validate the effectiveness of our approach.

---

### **Addressing the Major Concern on Long-Context Reasoning**

A major concern raised by several reviewers was the **lack of experiments on long-context reasoning tasks** (Reviewers **arKo**, **MbcZ**, **Dfk2**).

To address this, we added new experiments using **DeepSeek-r1-Distill-Llama-8B** on the **MATH-500** dataset, with output lengths ranging from **3K to 64K tokens**.

- These results have been added to the rebuttal and to the revised paper
  (**Sec. 4.3, Lines 404–421 & Fig. 5**).
- The new experiments demonstrate that our method reduces memory footprint for long-reasoning tasks and achieves **better performance under the same compute budget**.
- While the baseline supports only **6K context length with a score of 58.3**, our method scales to **64K** and achieves a score of **62.7** on the same computation node.

---

### **More revisions Based on Reviewer Suggestions**

We also incorporated several improvements into the updated manuscript:

1. **Simplified and clarified related work** (Reviewer **J7E7**, Sec 2).
2. **Clearer introduction** of the proposed approach and experimental setup (Reviewer **MbcZ**, Sec 3.3, 4.1).
3. **Additional analysis** explaining *why* our method outperforms baselines (Reviewer **arKo**, Sec 4.4).
4. **Discussion of cache-eviction strategies** and their effect on context-state updates (Reviewer **MbcZ**, Appendix C).

---

### **Discussion Stage Update**

During the discussion stage, **Reviewer MbcZ confirmed that our rebuttal resolved their concerns** and **raised the overall score** from 4 to **8** on **11/26 (morning, PST, one day before the reviewer-leak incident)**. The **average score** is improved from 5.0 to **6.0** correspondingly.

---

### **Closing Remarks**

Detailed answers to reviewer-specific questions can be found in the individual responses. We deeply appreciate all reviewers’ efforts and the valuable feedback that helped improve the clarity and quality of the paper.

We also sincerely thank the AC for their work.
**We hope this summary is helpful in your evaluation of our paper.**

---

### Meta-Review · Area_Chair_zW5Q · 2026-01-13

**Summary:**

Across reviewers, the paper was generally viewed as novel and practically motivated, with a clear contribution to improving the memory efficiency of RL-based post-training for reasoning models. Reviewers agreed that Progressive Thought Encoding is a clever and lightweight mechanism to mitigate KV-cache bottlenecks and that the experimental results consistently outperform LoRA and sliding-window baselines under constrained memory.

However, several substantive concerns initially limited enthusiasm:

- Insufficient long-context evaluation: Multiple reviewers (arKo, MbcZ, Dfk2) felt that the original 3K–6K token evaluations were not convincing for a method motivated by long-horizon reasoning.

- Model representativeness: Concerns were raised that some evaluated models (e.g., Qwen2.5-Instruct variants) are not true LRMs with long reasoning trajectories.

- Baseline fairness and clarity: Questions arose regarding fair comparison (e.g., use of global tokens in baselines), choice of LoRA as the primary baseline, and explanation of LoRA_c.

- Mechanistic clarity and capacity: Reviewers questioned whether a single global latent vector could realistically preserve long-range reasoning information and how eviction strategies affect learning.

- Presentation and analysis gaps: Several reviewers noted missing explanations for why the method works, incomplete related work, and clarity issues in method description.

The rebuttal and revision directly targeted these issues with extensive new experiments and clarifications, substantially strengthening the paper.

**Reviewer Concerns:**

Concerns Largely Addressed by the Rebuttal

- Long-context reasoning validity:
Addressed through new experiments up to 64K generation length on DeepSeek-R1-Distill-Llama-8B and additional results on DeepSeek-R1-Distill-Qwen-7B, demonstrating consistent gains under strict cache limits.

- Model representativeness:
Addressed by adding stronger LRM-style evaluations and clearly reframing Qwen models as evidence of generality rather than primary long-reasoning validation.

- Baseline fairness (global tokens, LoRA_c):
Addressed via new ablations incorporating global tokens into LoRA and LoRA_c, showing the proposed method’s gains persist beyond this factor.

- Mechanistic explanation:
Addressed with expanded analysis explaining how progressive encoding acts as a denoising/compression mechanism for evicted reasoning states.

- Eviction strategy sensitivity:
Addressed through new experiments and analysis showing how eviction granularity affects stability and performance.

- Applicability beyond LoRA-only setting:
Addressed with additional full-parameter RL fine-tuning experiments, showing the method remains competitive or superior.

- Presentation clarity:
Addressed by revising related work, clarifying terminology (e.g., LoRA_c, global query), and improving method exposition.

Concerns Partially or Still Outstanding

- Scalability to frontier-scale models (tens or hundreds of billions of parameters):
While discussed conceptually, this remains unvalidated empirically and is acknowledged as future work.

- Comparison with alternative architectural paradigms (e.g., hybrid state-space models):
The rebuttal provides a reasonable positioning argument, but no direct experimental comparison is included.

- Ultimate fidelity vs. full-cache reasoning at extreme lengths:
The authors convincingly show strong performance under constraints, but it remains clear (and acknowledged) that compressed representations cannot fully match unrestricted full-cache reasoning.

**Reviewer Scores:**

Reviewer J7E7:
Initially scored 6. Given the added fair baseline comparisons, clarification of global tokens, and revisions to related work, this reviewer would likely remain at 6.

Reviewer arKo:
Initially scored 6. The addition of long-context (up to 64K) experiments, DeepSeek-R1-Distill-Qwen-7B results, and deeper analysis directly address their main concerns so the reviewer would likely remain at 6.

Reviewer MbcZ:
Initially scored 4, explicitly stated that the rebuttal alleviated concerns and raised the score to 8 during discussion.

Reviewer Dfk2:
Initially scored 4. The rebuttal addressed baseline choice, infrastructure compatibility, long-context performance, and RL pipeline clarity. While some skepticism may remain, a reasonable expected increase would be to 6.

---

### Decision · Program_Chairs · 2026-01-26

Accept (Poster)